# Antifungal-Loaded Acrylic Bone Cement in the Treatment of Periprosthetic Hip and Knee Joint Infections: A Review

**DOI:** 10.3390/antibiotics11070879

**Published:** 2022-06-30

**Authors:** Konstantinos Anagnostakos, Sören L. Becker, Ismail Sahan

**Affiliations:** 1Zentrum für Orthopädie und Unfallchirurgie, Klinikum Saarbrücken, 66119 Saarbrücken, Germany; ssahan@klinikum-saarbruecken.de; 2Institut für Medizinische Mikrobiologie und Hygiene, Universitätsklinikum des Saarlandes, 66421 Homburg, Germany; soeren.becker@uks.eu

**Keywords:** bone cement, hip infection, knee infection, periprosthetic joint infection, *Candida* spp., antifungal-loaded bone cement

## Abstract

Little is known about the clinical use of antifungal-loaded acrylic bone cement in the treatment of periprosthetic hip and knee joint infections (PJIs). Hence, we performed a literature search using PubMed/MEDLINE from inception until December 2021. Search terms were “cement” in combination with 13 antifungal agents. A total of 10 published reports were identified, which described 11 patients and 12 joints in which antifungal-loaded cement was employed. All studies were case reports or case series, and no randomized controlled trials were identified. In 6 of 11 patients, predisposing comorbidities regarding the emergence of a fungal PJI were present. The majority of the studies reported on infections caused by *Candida* species. In six cases (seven joints), the cement was solely impregnated with an antifungal, but no antibiotic agent (amphotericin B, voriconazole, and fluconazole). In the other five joints, the cement was impregnated with both antibiotic(s) and antifungals. Great discrepancies were seen regarding the exact loading dose. Four studies investigated the local elution of antifungal agents in the early postoperative period and observed a local release of antifungals in vivo. We conclude that there is a paucity of data pertaining to the clinical use of antifungal-loaded bone cement, and no studies have assessed the clinical efficacy of such procedures. Future studies are urgently required to evaluate this use of antifungals in PJI.

## 1. Introduction

Periprosthetic joint infections (PJIs) after hip or knee arthroplasty are accepted to be a rare but hazardous complication with an incidence of 1–2% [1]. Several studies have tried to investigate the exact epidemiology and the microbiological profiles in either cohort studies [2,3,4,5] or from data gathered from national registries [6,7,8]. All studies agree that staphylococci are the most common causative organisms identified at the site of PJIs, whereas some geographical differences might be observed [1].

Fungal PJIs are considered a rarity among the identified organisms and represent less than 1% of the cases [9,10,11,12,13] (Figure 1). Although no clear guidelines regarding the management of fungal PJI exist [14], most studies recommend the two-stage procedure to be the treatment of choice [9,10,11,12,13]. Similar to bacterial infections, various treatment modalities including the use of spacers or beads or resection arthroplasty have been utilized [9,10,11,12,13].

Fungal PJIs are most frequently caused by yeasts (mainly *Candida* spp.) but may also be due to molds (e.g., *Aspergillus* spp.) and dimorphic fungi (e.g., *Coccidioides* spp.) [15]. As they are notoriously difficult to treat, attempts have been made over the past 15 years to investigate the effects of antifungal-loaded acrylic bone cement, with some promising results in vitro [16,17,18,19,20,21,22,23,24,25,26,27,28]. However, the clinical efficacy of antifungal-loaded bone cement is still unclear. Only few case reports have reported on this topic [29,30,31,32,33,34].

Clinical studies [9,11] but also reviews [10,12,13] about fungal PJIs mostly concentrate on the surgical treatment and the clinical or infectiological outcome. To the best of our knowledge, a literature review about the role of the local antimycotic therapy by means of antifungal-loaded cement in the treatment of hip and knee PJIs has not been published, yet. Here, we want to summarize this knowledge, and draw conclusions about this treatment option. 

## 2. Results

Using a PubMed/MEDLINE search from inception to 31.12.2021, a total of 11 studies could be identified (Figure 2) [29,30,31,32,33,34,35,36,37,38,39]. One study had to be excluded from the evaluation process, because the relevant data could not be clearly extracted from the article [29]. 

The remaining 10 studies [29,30,31,32,33,34,35,36,37,38] reported on 11 clinical cases (12 joints; one patient suffered from a bilateral fungal PJI [34]). No study was published before 1998. All studies had a level of evidence IV (i.e., case series, low-quality cohort, or case–control studies). In half of the studies, predisposing comorbidities regarding the emergence of a fungal PJI were present. The great majority of the studies reported on infections caused by *Candida* species, whereas in one case a coexisting bacterial infection was detected. All data about demographic information, predisposing comorbidities, and fungal organisms are summarized in Table 1.

### 2.1. Cement Loading and Pharmacokinetic Properties

In three studies, the cement used was Palacos^®^ and in one Simplex^®^. The remaining six studies did not provide any information about the type of cement used. In seven joints, the cement was solely impregnated with an antifungal agent (amphotericin B, voriconazole and fluconazole in two studies, respectively). In the other five joints, the cement was impregnated with both antibiotic(s) and antifungals. However, great discrepancies were present with regard to the exact loading dose. Some studies only stated the total amount inserted into the cement, whereas other mentioned the amount per package cement. All data are summarized in Table 2.

Four studies investigated the local elution of antifungal agents in the early postoperative period. Deelstra et al. determined the serum and drain fluid levels of voriconazole and amphotericin B 24, 48, and 72 h after surgery [30]. All the elution levels of both agents were above the minimum inhibitory (MIC) levels determined for *C. albicans*, as derived from recommendations put forth by the National Committee for Clinical Laboratory Standards M27–A2. Marra et al. similarly measured the amphotericin B concentrations in blood and drain fluid [33]. Blood samples were collected at 3, 6, 12, 26, 38, 50, 69, 93, 117, 141, 165, 189, and 213 h after the spacer insertion. Drainage fluid specimens were obtained at 6, 12, 26, 50, 74, 98, and 122 h after the surgery. The serum concentration reached a peak of 1.2 mg/L at 6 h after the implantation, and then continuously declined, until 50 h after implantation no levels of amphotericin B could be detected. In the drainage fluid, a maximum concentration of 3.2 mg/L was determined at 50 h. Over the next 24 h, this concentration steadily declined and reached a plateau over the following 48 h (0.57 and 0.51 mg/L). Denes et al. treated a 55-year-old male patient suffering from a bilateral hip PJI by bilateral implantation of voriconazole-loaded spacers [35]. Two weeks after the left and 1 week after the right hip surgery, both hips were aspirated. The voriconazole concentrations were 0.04 mg/L in the left and 0.1 mg/L in the right hip. Bruce et al. treated two cases of hip PJI by insertion of fluconazole-loaded cement beads [37]. After the first stage, the levels of fluconazole were measured in both cases in the deep and superficial drain fluid, respectively. After initial high levels, the concentrations were maintained at about 5 mg/L for several days. The total amount of fluconazole released was 21 mg in the first and 13.9 mg in the second case after 10 and 7 days, respectively.

### 2.2. Surgical Treatment, Systemic Therapy, and Outcome

In 9 out of 10 studies, the patients were treated by a two-stage procedure. All studies except for one reported about antifungal-loaded cement spacers. In this one study, beads were used instead of a spacer [37]. The time period between stages ranged between 2 and 10 months (Table 2).

In the majority of the studies, fluconazole was the agent of choice for the systemic therapy, followed by voriconazole. The length of therapy varied from 6 to 20 weeks, whereas Reddy et al. continued the therapy for 10 additional weeks (total length of therapy 30 weeks) after prosthesis reimplantation [36].

At prosthesis reimplantation, two studies reported on the use of antifungal-loaded cement for fixation of the prosthesis. Bruce et al. used fluconazole-loaded cement in one out of two cases [36], whereas Wu et al. impregnated the cement with vancomycin and amphotericin B [31].

Complications were seen in two studies. Gaston and Ogden reported on a culture-negative PJI at the time of the second stage, ending in an above-knee amputation [32]. Marra et al. described a reinfection with *Escherichia coli* at reimplantation [33]. In seven studies, no complications were observed (Table 2).

## 3. Discussion

Since the first description of a *Candida* PJI case in 1979 by McGregor et al. [40], fungal PJIs have gained an increasing interest, especially in the past 10 years [9,10,11,12,13]. Generally, risk factors for the development of invasive candida infections include immunosuppression, neutropenia, chronic, or prolonged use of antibiotics, presence of indwelling catheters, parenteral hyperalimentation, malnutrition, diabetes mellitus, rheumatoid arthritis, cirrhosis, history of abdominal surgeries, history of renal transplantation, severe burns, and injection drug use [38]. HIV infection, neutropenia due to chemotherapy, or diabetes mellitus lead to increased susceptibility of cutaneous infection with *Candida* and possibly other fungal pathogens, e.g., dermatophytes [31]. Multiple surgeries, the repetition of extended courses of empiric antibiotic, and the presence of indwelling catheters might act as a predisposing factor for the emergence of a fungal PJI [34]. Three possible etiologies of fungal PJI have been proposed, including direct seeding as in trauma, iatrogenic (surgery), and hematogenous spread [30,31]. Despite knowledge regarding all these risk factors, approximately one-half of the reported cases of fungal PJI have no identifiable risk factor [38], which was also observed by our findings.

### 3.1. Surgical Treatment Recommendations

Based on the fact that no strict guidelines about the management of fungal PJIs exist, current literature shows that the surgical treatment options strongly vary. A major problem is hereby that clinical symptoms are often mild, and the correct diagnosis is often delayed, especially when no selective fungal media have been used for microbiological examination [13]. Moreover, radiological signs of prosthesis loosening are only evident in approximately 50% of the cases [11]. Hence, the treatment of these rare infections still remains challenging.

A systematic review of fungal PJI in total knee arthroplasty involving 45 cases from 36 included studies demonstrated very heterogenous surgical treatment options including two-stage arthroplasty, permanent resection arthroplasty, delayed reimplantation, and delayed arthrodesis [13]. A similar systematic review about fungal PJIs of the hip (45 cases–21 studies) revealed an equal heterogeneity among the treatment options, consisting of one- or two-stage exchange, debridement, antibiotics, irrigation, and retention of the prosthesis (DAIR), and chronic suppression therapy with an antimycotic agent [12].

In the last International Consensus Meeting on Musculoskeletal Infections in 2018, an attempt was made to summarize the current literature about diagnosis and treatment of these rare infections [41]. Although high-level evidence is limited, a recommendation was made that two-stage procedures remain the treatment of choice. DAIR procedures should be only limited to those cases with early presentation, good soft-tissue coverage, well-fixed implants, and that are healthy patients. Regarding the systemic antifungal therapy, this should be continued for up to 6 months or longer, if necessary [41]. 

Based on these data, our clinical experience with fungal PJIs and the fact that most patients present rather with a chronic infection due to the delayed diagnosis than within few days after the onset of their symptoms, we favor the two-stage option as the treatment of choice. Prior to the beginning of the surgical revision, specific attention should be paid for an accurate and timely diagnosis in order to avoid any further unnecessary delays. 

### 3.2. Antifungal-Loaded Bone Cement

Antibiotic-loaded bone cement is an established tool in the treatment of hip and knee PJIs. At the site of a two-stage protocol by means of either spacer or beads implantation, infection eradication rates of more than 90% are reported [42]. The locally released antibiotic concentrations exceed by far those after systemic administration at being systemically safe, making it therefore attractive from a microbiological point of view [42]. The antibiotic elution itself is defined by two different phases: initially, a very high release occurs within the first 7–10 days, followed by a much slower elution that constantly declines over a time period of four to six weeks. Moreover, the antibiotic elution is influenced by various parameters, such as type of cement used, type and number of antibiotic agents (which might lead to a synergistic effect between the antibiotics or not) and the impregnation dosage [42]. Therefore, a broad knowledge about all this information is an indispensable premise for a successful clinical outcome and prevention of persistence of infection. 

Despite the wide clinical use of antibiotic-loaded bone cement, the role of antifungal-impregnated bone cement is still controversial. The optimal characteristics of antifungal agents for incorporation into bone cement have not been defined yet. It could be probably assumed that these characteristics should be similar to those of antibiotics (availability in powder form, wide antifungal spectrum, fungicidal at low concentrations, elution from bone cement in high concentrations for prolonged periods, thermal stability, low or no risk of allergy or delayed hypersensitivity, low influence on the mechanical properties of the cement and a low serum protein binding) [42].

#### 3.2.1. Antifungal-Loaded Bone Cement—In Vitro and Animal Studies

Most information about antifungal-loaded bone cement originates from experimental studies. Amphotericin B [16,17,18,19,20,21,22,23,24], voriconazole [25,26,27,43], and fluconazole [22,24] have been the most common investigated agents. Silverberg et al. showed that fluconazole and amphotericin B remained active after cement polymerization, whereas fluorcytosine did not [24]. The release of the antifungal agents was higher from Palacos^®^ than from Simplex^®^ cement. Regarding the use of amphotericin B, Goss et al. described poor elution characteristics with a cumulative release rate of less than 0.03% from the incorporated amount [23]. On the other side, Houdek et al. reported that sufficient amounts of deoxycholate amphotericin B are released from beads in vitro [17]. Cunningham et al. found that the use of liposomal amphotericin B leads to a higher release from bone cement compared with the use of deoxycholate amphotericin B [20]. These findings were later confirmed by the study of Czuban et al. [16]. This liposomal formulation of amphotericin B was found to be safe in a mouse model, although it was cytotoxic under in vitro circumstances [18]. Kweon et al. showed that the addition of high-dose poragen leads to an increase from amphotericin B, but compressive strength decreases sufficiently to limit its use for implant fixation [21]. In a continuous flow chamber model, Rouse et al. found that after cement polymerization the amount of voriconazole was 5.6% compared with anidulafungin with 0.7% [43]. Peak concentrations of voriconazole reached 0.9 µg/mL, whereas anidulafungin was not detected at all, indicating that it was not eluted from bone cement. Grimsrud et al. demonstrated that voriconazole can be released in biologically active concentrations over a time period of at least 2 weeks [25]. Miller et al. reported that the elution of voriconazole is dose-dependent, and these concentrations can be detected over at least 30 days [27].

#### 3.2.2. Antifungal-Loaded Bone Cement—Clinical Studies

Based on these in vitro data, it is apparent that a limited amount of experience exists only for fluconazole, voriconazole, and amphotericin B, which can be therefore regarded as potential agents for impregnation of bone cement. Nevertheless, it should be born in mind that data and conclusions from in vitro studies cannot be directly converted to those from in vivo studies. The type, amount, and ratio of the used agent(s), the type and porosity of cement, the surface characteristics, the way the cement is prepared, and the environmental circumstances are accepted to be factors with a possible affection on the elution kinetics from bone cement [44]. Therefore, there is no consensus regarding the ideal type and dose of antifungal agents that can be used in clinical practice against PJI. Pharmacokinetics, safety, published reports, drug interactions, and isolate susceptibility must be considered when selecting a therapy [14]. This is also confirmed by our findings. Only these three aforementioned antifungals have been used in clinical practice, whereas no universal impregnation dosage could be observed for each single agent (Table 2). Apparently, most authors orientated to the usual daily dosage of each drug, whereas it is unclear whether such a dosage accounts for an optimal cement impregnation and hence antifungal elution. This topic requires further investigation. 

As identified here, only four studies investigated the local elution of antifungal agents in the early postoperative period. A positive finding was that antifungals can be released from bone cement not only in vitro but also in vivo. Denes et al. proved that voriconazole had remained stable and detectable despite the exothermic reaction of cement polymerization [35]. Based on their results, Marra et al. concluded that amphotericin B added to bone cement will result in diffusion of the drug to blood and local wound fluid [33]. However, the height of the locally determined concentrations was lower compared with those measured at the site of antibiotic-loaded hip or knee spacers [44]. Nonetheless, it is questionable whether antifungals have to be locally released in such high amounts since the MIC of fungal organisms is different than that of bacteria.

The ideal impregnation of antifungal agents should be further investigated in future studies. An inappropriate or suboptimal impregnation of cement beads or spacers might have a negative impact on the clinical course and the eradication of the joint infection. Should any fungal organisms have survived the surgical debridement and if these organisms develop resistances due to locally released subinhibitory concentrations, the risk of persistence of the infection might dramatically rise, especially since it is known that biofilms do form in fungal infections similar to the bacterial infections. An in vitro study on fungal biofilm has shown that *C. albicans* is producing quantitatively more biofilm than *Candida parapsilosis*, *Candida glabrata,* and *Candida tropicalis*. These films have rapidly developed resistance against fluconazole [45]. Moreover, it is unclear whether the bone cement should be additionally loaded with antibiotics. Some authors recommend the additional impregnation of bone cement with antibiotics, since a subsequent bacterial infection onto a present fungal one is a known complication at the site of these infections [46], which was also described by Marra et al. [33] in our findings.

Interestingly, two studies reported on the use of antifungal-impregnated cement for prosthesis fixation at prosthesis reimplantation [31,37]. It is unclear how the addition of antifungals to the cement might influence the long-term stability of the prosthesis. Although the authors of these studies did not report on any prosthesis loosening one year [31] and seven years [37] after reimplantation, respectively, the limited numbers of cases and the length of follow-up do not allow for generalization of conclusions. Therefore, this procedure cannot be routinely recommended.

### 3.3. Systemic Antifungal Therapy

The systemic antifungal therapy is also of importance, although the choice and duration of the antifungal therapy also remain controversial. At the site of a septic fungal arthritis, the Infectious Diseases Society of America (IDSA) recommends either 400 mg fluconazole daily for 6 weeks or an echinocandin (e.g., caspofungin) for 2 weeks followed by 400 mg fluconazole for at least 4 weeks [14]. The lipid formulation of amphotericin B for 2 weeks, followed by fluconazole for at least 4 weeks is a less attractive alternative. If the prosthetic device cannot be removed, chronic suppression with fluconazole, if the isolate is susceptible, is recommended [14]. The absence of serious adverse effects and favorable pharmacokinetic features of rapid oral absorption with high bioavailability, extended half-life allowing once daily administration, and high concentration of fluconazole in joint fluid approximating that in plasma [47], make fluconazole a good choice for the treatment of fungal PJI, especially from an orthopedic point of view [31].

Conventional amphotericin B is one of the most toxic antiinfective drugs. Common side effects of parenteral administration include immediate infusion-related reactions of hyperpyrexia, severe malaise and hypotension, acute renal failure, anemia, hypokaliemia and occasional leukopenia and thrombocytopenia [33]. The introduction of liposomal formulations of amphotericin B have dramatically reduced the incidence of severe side effects. On the other hand, although rare, resistance of *Candida* species to azole drugs has been reported [39]. Similar confusion regards the duration of the systemic antifungal therapy. According to some authors, patients should be treated for up to 4 weeks after resolution of clinical signs and symptoms of infection and when there is microbiological evidence of infection eradication. Others recommend that therapy should continue for at least 6–12 months beyond the resolution of clinical symptoms and signs of infection [9,48]. Fluconazole was the agent of choice in most studies, whereas the length of therapy ranged from 6 to 30 weeks.

Similar to the length of the systemic antifungal therapy, the time period between the surgeries at the site of a two-stage procedure ranged between 2 and 10 months. This raises the question whether antifungal-loaded bone cement in combination with a prolonged systemic therapy might lead to any adverse systemic reactions. Antibiotic-loaded bone cement is regarded to be systemically safe, although single reports have emphasized potential side effects such as acute renal failure, dermatologic reactions, bone marrow depression, or hepatic failure [49]. Whether this possible complication also accounts for antifungal-loaded bone cement remains unclear and cannot be answered based on the small number of studies evaluated in the present review.

Our study has some limitations. In accordance with the inclusion criteria, only 10 clinical studies could be identified and evaluated, which makes a statistical evaluation not possible. The literature search was performed only through one database. We cannot exclude the fact that a search through more databases might have led to the identification of a higher number of relevant studies. Moreover, we evaluated only studies that reported on the treatment of hip and knee PJIs and did not include other entities with a fungal infection (e.g., osteomyelitis) or other localizations.

## 4. Materials and Methods 

A literature search was performed through PubMed/MEDLINE until 31 December 2021 (Figure 1). Search terms were “cement” and 13 antifungal agents that can be systemically applied (amphotericin B, voriconazole, ketoconazole, itraconazole, fluconazole, posaconazole, isavuconazole, fluorocytosine, caspofungin, griseofulvine, terbinafine, anidulafungin, and ciclopirox). Only English studies were included. Reviews were excluded. Only studies that solely reported on the clinical use of an antifungal-loaded bone cement device (beads and spacers) in the treatment of hip and knee PJIs were included. In vitro studies or those that described the use of such device in other localizations or indications were excluded. Among the primarily identified studies, a search was carried out through the bibliography of each article for further identification of relevant studies. All publications were analyzed with regard to publication year, level of evidence, number of joints/patients treated, demographic data (age, gender, comorbidities), causative fungal organism, cement type and impregnation, surgical treatment, type and length of systemic therapy, complications, and length of follow-up. 

## 5. Conclusions

In conclusion, literature data are scarce about the clinical use of antifungal-impregnated bone cement in the management of fungal hip and knee PJIs. Single studies provide information that different antifungal agents can be locally released under in vivo circumstances. However, the ideal impregnation type and dose of antifungals has not been defined, yet. We were able to find a single source providing a recommendation about the impregnation of bone cement with antifungal agents (Pro-Implant Foundation, Berlin, Germany) [50], whereas it is unclear on which data this recommendation is based. Therefore, future studies should investigate the optimal impregnation of bone cement with antifungal agents and evaluate its clinical use in larger collectives. Such a study might be most likely feasible in a multi-center design and take into consideration not only the impregnation dosage of the antifungal agents but also the length of the systemic therapy.

## Figures and Tables

**Figure 1 antibiotics-11-00879-f001:**
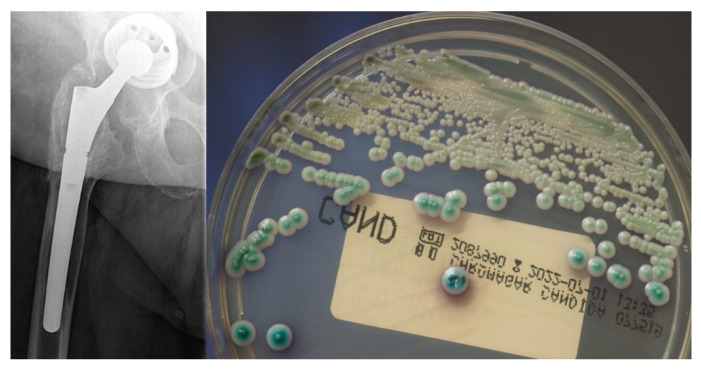
Left: Anterio-posterior X-rays of the right hip of a 66-year old male patient with a loosening of the acetabular cup and large osteolyses; right: the microbiological examination revealed growth of a *Candida* spp. isolate (here on a chromogenic agar medium).

**Figure 2 antibiotics-11-00879-f002:**
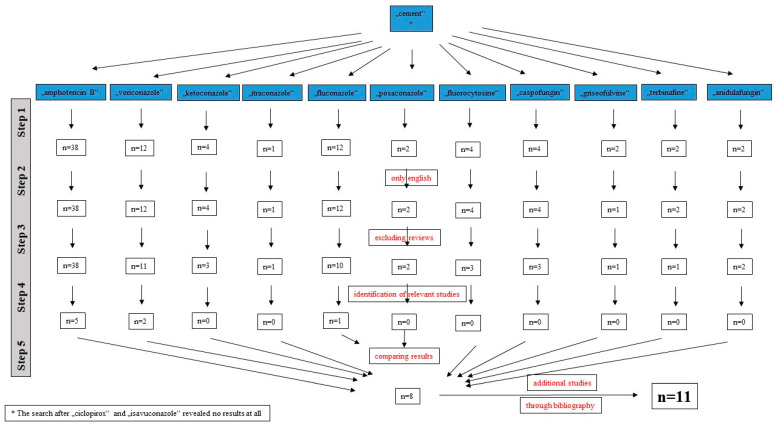
Literature search for identification of studies about the use of antifungal-loaded acrylic bone cement in the treatment of periprosthetic hip and knee joint infections.

**Table 1 antibiotics-11-00879-t001:** Overview of treated cases, demographic data, predisposing comorbidities, and causative organism(s).

Study	Publication Year	Joint	No. of Cases/Joints	Age	Sex	Predisposing Comorbidities	Fungal Organism
Deelstra et al. [30]	2013	Hip	1/1	73 y.	Female	None	*Candida albicans* + coagulase-negative staphylococci
Wu et al. [31]	2011	Knee	1/1	72 y.	Male	None	*Candida albicans*
Gaston and Ogden [32]	2004	Knee	1/1	42 y.	Female	Steroids for lupus	*Candida glabrata*
Marra et al. [33]	2001	Hip	1/1	59 y.	Male	2 × aseptic revision arthroplasty surgeries	*Candida albicans*
Burgo et al. [34]	2017	Hip	1/1	53 y.	Female	Corticosteroids for myasthenia gravis, chronic hepatitis B, obesity, non-insulin-dependent diabetes mellitus, prior PJI	*Trichosporon inkin*
Denes et al. [35]	2012	Hip	1/2	55 y.	Male	n.r.	*Candida glabrata*
Reddy et al. [36]	2013	Knee	1/1	62 y.	Female	None	*Candida tropicalis*
Bruce et al. [37]	2001	Hip	2/2	51 y./ 68 y.	Female/Female	n.r.	*Candida parapsilosis*, *Candida albicans*
Phelan et al. [38]	2002	Hip	1/1	75 y.	Female	Rheumatoid arthritis, prior PJI	*Candida albicans*
Selmon et al. [39]	1998	Knee	1/1	75 y.	Female	Prior abdominal surgery	*Candida glabrata*

n.r.: not reported; PJI: periprosthetic joint infection.

**Table 2 antibiotics-11-00879-t002:** Data about the use of antifungal-loaded acrylic bone cement and treatment modalities.

Study	Cement Used	Cement Impregnation	Surgical Treatment	2-Stage Interval	Systemic Therapy	Length of Systemic Therapy	Complications	Follow-Up
Deelstra et al. [30]	Palacos	0.5 g gentamicin + 1 g vancomycin + 1 g voriconazole + 0.25 g amphotericin B/40 g cement	2-stage	3 months	Fluconazole + vancomycin	n.r.	None	6 years
Wu et al. [31]	n.r.	1.2 g amphotericin B/40 g cement	2-stage	6 months	Fluconazole	6 months	None	1 year
Gaston and Ogden [32]	n.r.	Vancomycin + amphotericin B	2-stage	2 months	Voriconazole	2 months	Culture-negative infection after 2 months, ending in an above-knee amputation	6 months
Marra et al. [33]	Palacos	Total 750 mg amphotericin B	2-stage	10 weeks	Fluconazole	6 weeks	Reinfection with *E. coli* at reimplantation	n.r.
Burgo et al. [34]	n.r.	Voriconazole + vancomycin	2-stage	n.r.	Voriconazole	6 months	None	2 years
Denes et al. [35]	Simplex	Total 600 mg voriconazole right hip/total 400 mg voriconazole left hip	2-stage	n.r.	Caspofungin	n.r.	n.r.	n.r.
Reddy et al. [36]	n.r.	Vancomycin + amphotericin B	2-stage	20 weeks	Fluconazole	18 weeks	None	2 years
Bruce et al. [37]	Palacos	2 g fluconazole/ 40 g cement	2-stage	10 months/ 3 months	Fluconazole	n.r.	None	7 years/ 4 years
Phelan et al. [38]	n.r.	Total 200 mg fluconazole	2-stage	2.4 months	Fluconazole	47 days	None	17 months
Selmon et al. [39]	n.r.	Gentamicin + 200 mg amphotericin B	1-stage	n.r.	Amphotericin B/itraconazole	1 week/ 8 weeks	None	4 years

n.r.: not reported; CNS: coagulase-negative staphylococci.

## Data Availability

All relevant data are presented in the article.

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
