# Peer review of "Antifungal-Loaded Acrylic Bone Cement in the Treatment of Periprosthetic Hip and Knee Joint Infections: A Review"

_antibiotics, 2022, doi:10.3390/antibiotics11070879_

Round 1

Reviewer 1 Report

Dear Anagnostakos et al.,

The manuscript “review the current literature about the use of antifungal-loaded cement in the treatment of hip and knee PJIs” (antibiotics-1752284) by Anagnostakos et al. aims to review the current literature about the use of antifungal-loaded cement in the treatment of hip and knee PJIs. The topic is interesting, but I think this article should reconsider after proper changes in major revision for publication in Antibiotics. Some of my specific comments are below:

1. I found a strange sentence in line 52-53, “…no study has tried to review the current 52 literature about…”. How study/research perform review? The authors seem not to know about the original research article and review article as the present sentence in this article. Please revise it.

2. The state of the art and the significance of the current review are not clearly present, the authors should highlight it more advanced in the introduction section (line 31-55).

3. It is needed to add at least one illustrative figure in the introduction section to make the reader more interested and easier to understand in the present review.

4. Since this manuscript discusses regarding bone cement on periprosthetic hip and knee joint, I would encourage and advise the authors to adopt some of the specific additional references related to total hip prosthesis bearing published by MDPI in the introduction section (line 34-85) as follow:

Tresca Stress Simulation of Metal-on-Metal Total Hip Arthroplasty during Normal Walking Activity. Materials (Basel). 2021, 14, 7554. https://doi.org/10.3390/ma14247554

5. I do not understand for the present author's manuscript if after introduction section in the review article is followed by results section. Before the results section, it should behave a methods section. Please revise it.

6. In line 57-58, it is not valid if only one main source of database like Pubmed/MEDLINE. The authors are recommended to include Scopus and Web of Science on performing search literature.

7. The quality of figure 1 (line 62-64) is too poor, please improve the image quality and make it colored, not only black color.

8. Why materials and methods is in section 4 (line 227-239)? It should be in section 2. It is sos strange and should be revised.

9. The limitation of the present article needs to be included in the discussion section.

10. The conclusion section is missing and should be provided. I see the authors have a conclusion in line 220-226, but it should be written in a separate section. Also, more elaboration is needed.

11. Further research needs to be explained in the conclusion section.

12. Overall, I do not see a serious scientific contribution bringing in the existence of this review article. I encourage the authors to add major substance to the present review article. It is not solid and should be improved in quality and quantity.

13. I see some errors on English in some areas of the present manuscript. To improve the quality of English used in this manuscript and make sure English language, grammar, punctuation, spelling, and overall style are correct, further proofreading is needed. As an alternative, the authors can use the MDPI English proofreading service for this issue.

14. Please make sure the authors have used the Antibiotics, MDPI format correctly. The authors can download published manuscripts by Antibiotics, MDPI, and compare them with the present author's manuscript to ensure typesetting is appropriate. For example:

Email of the authors should be written with black color without underline, see line 9 and revise it

Uppercase and lowercase of the title and all of the subsection is not appropriate based on the format

And other

I am pleased to have been able to review the author's present manuscript. Hopefully, the author can revise the current manuscript as well as possible so that it becomes even better. Good luck for the author's work and effort.

Best regards,

The Reviewer

Author Response

Dear Reviewer,

first of all, we would like to thank you for reviewing our manuscript and all the comments you made.

We have thoroughly read all your comments and made all revisions as followed (all revisions are highlightened red in the text)

The manuscript “review the current literature about the use of antifungal-loaded cement in the treatment of hip and knee PJIs” (antibiotics-1752284) by Anagnostakos et al. aims to review the current literature about the use of antifungal-loaded cement in the treatment of hip and knee PJIs. The topic is interesting, but I think this article should reconsider after proper changes in major revision for publication in Antibiotics. Some of my specific comments are below:

  1. I found a strange sentence in line 52-53, “…no study has tried to review the current 52 literature about…”. How study/research perform review? The authors seem not to know about the original research article and review article as the present sentence in this article. Please revise it.

Response: Thank you for your comment. Previous original works but also reviews concentrated rather on the surgical treatment and the clinical outcome i.e. infection outcome than to investigate the possible effect of the local antimycotic therapy. Therefore, we have stated that no study has investigated this topic. We have revised this part in order to make clearer for the readers (Introduction, page 2, lines 57-61).

  1. The state of the art and the significance of the current review are not clearly present, the authors should highlight it more advanced in the introduction section (line 31-55).

Response: Similar to our response to your first comment, we have revised the end of the Introduction in order to clarify this point (Introduction, page 2, lines 57-61).

  1. It is needed to add at least one illustrative figure in the introduction section to make the reader more interested and easier to understand in the present review.

Response: We thank the reviewer for this suggestion. While revising, we have added one figure of a microbiological culture growing a Candida spp. isolate on a chromogenic agar (Introduction, page 2).

  1. Since this manuscript discusses regarding bone cement on periprosthetic hip and knee joint, I would encourage and advise the authors to adopt some of the specific additional references related to total hip prosthesis bearing published by MDPI in the introduction section (line 34-85) as follow:

Tresca Stress Simulation of Metal-on-Metal Total Hip Arthroplasty during Normal Walking Activity. Materials (Basel). 2021, 14, 7554. https://doi.org/10.3390/ma14247554

Response: Thank you for your proposal. We have checked the suggested work, but we found it not suitable for citation in the present work. Therefore, we did not change anything here. We hope you are fine with that.

  1. I do not understand for the present author's manuscript if after introduction section in the review article is followed by results section. Before the results section, it should behave a methods section. Please revise it.

Response: Thank you for your comment. We understand that it is unusual that after the Introduction section the Results section appears. However, we used the manuscript template provided by Antibiotics (Basel) and this order of the sections is provided. We have published last year two articles in this journal and it was the same. Therefore, we did not change anything here.

  1. In line 57-58, it is not valid if only one main source of database like Pubmed/MEDLINE. The authors are recommended to include Scopus and Web of Science on performing search literature.

Response: Thank you for your comment. There exist no specific recommendations according to which databases are defined or recommended to use for literature reviews. In literature and independent on the topic, some reviews are based on the results of a single database, whereas other use several ones. However, you are right with the fact that if we had searched through more databases a different number of relevant studies might have been identified for inclusion into the study. We have added this point to the limitation of the study in the end of the Discussion section (lines 255-262), however, we did not perform a new additional literature search including other databases. We hope you agree with that.

  1. The quality of figure 1 (line 62-64) is too poor, please improve the image quality and make it colored, not only black color.

Response: We have changed it accordingly.

  1. Why materials and methods is in section 4 (line 227-239)? It should be in section 2. It is sos strange and should be revised.

Response: As we afore mentioned under 5., we used the manuscript template provided by Antibiotics (Basel) and this order of the sections is provided. We have published last year two articles in this journal and it was the same. Therefore, we did not change anything here.

  1. The limitation of the present article needs to be included in the discussion section.

Response: Thank you for your comment. We have added a paragraph about the limitations of the study in the end of the Discussion (lines 255-262).

  1. The conclusion section is missing and should be provided. I see the authors have a conclusion in line 220-226, but it should be written in a separate section. Also, more elaboration is needed.

Response: Thank you for your comment. We have added a separate Conclusions section (section 5, lines 281-294).

  1. Further research needs to be explained in the conclusion section.

Response: As suggested, we added some information about this topic (lines 281-294).

  1. Overall, I do not see a serious scientific contribution bringing in the existence of this review article. I encourage the authors to add major substance to the present review article. It is not solid and should be improved in quality and quantity.

Response: Thank you for your comment. The treatment of periprosthetic joint infections usually consist of three columns: the surgical management, the systemic antibiotic/antimycotic therapy, and the local antibiotic/antimycotic therapy. As we have pointed out in the manuscript, no study has tried to review the current literature about the local antfungal therapy in the treament of hip and knee PJIs. We represent the opinion that such reviews pose an important tool in the dissemination of medical knowledge and therefore are a scientific contribution to this field. By making all revisions suggested by you and the other reviewers, we believe that the quality and the quantity of our work is improved.

  1. I see some errors on English in some areas of the present manuscript. To improve the quality of English used in this manuscript and make sure English language, grammar, punctuation, spelling, and overall style are correct, further proofreading is needed. As an alternative, the authors can use the MDPI English proofreading service for this issue.

Response: Our study group has published numerous english articles and we had never have any language issues. However, if you you believe that our work needs any language improvement, this could be initiated within the proofreading proccess.

  1. Please make sure the authors have used the Antibiotics, MDPI format correctly. The authors can download published manuscripts by Antibiotics, MDPI, and compare them with the present author's manuscript to ensure typesetting is appropriate. For example:

Email of the authors should be written with black color without underline, see line 9 and revise it

Uppercase and lowercase of the title and all of the subsection is not appropriate based on the format

And other

Response: Thank you for your comments, we have revised the title of the manuscript, those of the subsections and the email accordingly.

Best regards

The authors

Reviewer 2 Report

In this review by Anagnostakos et al., the authors performed a literature search in Pubmed/Medline for studies related to antifungal containing cements used in prosthetic hip and knee joint infection (PJI). They focused on 10 such reports and discussed the status of antifungal usage, the agent-containing cement materials, and the elution of the agents from the cements. There are some suggestions/comments that need to be addressed.

11.   The review will benefit from discussion about the challenges of fungal infection treatment in PJI. The authors can also include their opinion about the ways in which the treatment can be improved.

22.    The study inclusion criteria should be explained better. It is not clear why the authors ended up focusing only on 10 out of the total 75 studies. How did they screen for the “relevant studies”?

33.     Page 3 of 15, line 35-36 is not clear. Is it the fluconazole amount released over a period or concentration?

Author Response

Dear Reviewer,

first of all, we would like to thank you for reviewing our manuscript and all the comments you made.

We have thoroughly read all your comments and made all revisions as followed (all revisions are highlightened red in the text)

  1.  The review will benefit from discussion about the challenges of fungal infection treatment in PJI. The authors can also include their opinion about the ways in which the treatment can be improved.

Response: Thank you for your comment. As suggested, we have added these information in the begin of the Discussion under the new section 3.1. (surgical treatment recommendations, lines 84-101, and lines 111-116).

  1.   The study inclusion criteria should be explained better. It is not clear why the authors ended up focusing only on 10 out of the total 75 studies. How did they screen for the “relevant studies”?

Response: Thank you for your comment. We have specified this unclear point and added this information in the Materials and Methods section (lines 269-273).

  1. Page 3 of 15, line 35-36 is not clear. Is it the fluconazole amount released over a period or concentration?

Response: Thank you for the question, These amounts were released after 10 and seven days, respectively. The missing information has been added to the text.

Best regards

The authors

Reviewer 3 Report

An article with great potential, but with few statistics relevant to the magnitude of the subject. I recommend the inclusion of several relevant studies in the field, in order to have a statistically significant representation.

I recommend an adjustment of the organization of the article, in a more coherent and practical form.

Author Response

Dear Reviewer,

first of all, we would like to thank you for reviewing our manuscript and all the comments you made.

We have thoroughly read all your comments and made all revisions as followed (all revisions are highlightened red in the text)

You are correct with the fact that no „true“ statistically significant representation has been made. To be honest, it is scientifically very difficult, if not impossible, to perform a statistical evaluation when only 10 studies were found to be relevant. This is certainly a limitation of the present work, and we have pointed out in a new paragraph in the end of the Discussion section. We do not regard any additional inclusion of further studies to be productive, only with the aim of a statistical evaluation, because we would be forced to compare apples with pears (e.g. if we would expand the indications and localizations where antifungal-loaded bone cement device have been clinically used).

However, we agree with you that a review article should be presented in a coherent and practical form that makes it easy to read and understand. Based on the comments from the other reviewers, several revisions were made through the whole text. We re-organized the Discussion section, added subchapters and additional information in the management of antifungal PJIs. By that, we hope that the quality of our work has been improved.

Best regards

The authors

Round 2

Reviewer 1 Report

Appreciate the author's effort in their revision after peer review. Authors have made an effort to respond to every comment that has been criticized. Unfortunately, the writer failed to understand the points conveyed by the reviewer in the previous report.

In the first comment, this manuscript becomes ambiguous between its position as a review article and also a research article that the author failed to explain. On the second point, the significance of this review is still unclear after the revision. On the third point, the addition of figure 1 can help, but it seems inappropriate because it does not explain any information constructively. The fourth point related to the explanation of cement on periprosthetic hip and knee joints also failed to be carried out by the author. The fifth point, after the introduction section, is not followed by the results, but the method, protocol, or procedure. On the sixth point, the use of a single database makes the literature not rich so it is weak in discussion making this work not have a strong scientific contribution. The following points are also not addressed by the author precision.

Because serious flaws in the current article make it dangerous to the scientific community and its low-quality means that this article should be rejected.

Author Response

Appreciate the author's effort in their revision after peer review. Authors have made an effort to respond to every comment that has been criticized. Unfortunately, the writer failed to understand the points conveyed by the reviewer in the previous report.

Response: 

Dear Reviewer,

thank you again for reviewing our manuscript and all your comments.

As with every review, we take all criticism very serious because we understand that by that each reviewer tries to contribute to the scientific paper and makes it better. Below, we would like to reply to each of your points

In the first comment, this manuscript becomes ambiguous between its position as a review article and also a research article that the author failed to explain. On the second point, the significance of this review is still unclear after the revision. On the third point, the addition of figure 1 can help, but it seems inappropriate because it does not explain any information constructively. The fourth point related to the explanation of cement on periprosthetic hip and knee joints also failed to be carried out by the author. The fifth point, after the introduction section, is not followed by the results, but the method, protocol, or procedure. On the sixth point, the use of a single database makes the literature not rich so it is weak in discussion making this work not have a strong scientific contribution. The following points are also not addressed by the author precision.

Response: 

  • First point: as we mentioned in our last response, we have added an explanation in the end of the Introduction (lines 56-61), pointing out that no article has tried to summarize the literature data about the clinical use of antifungal-loaded bone cement in the treatment of periprosthetic hip and knee joint infections. By that, the reader gets the opportunity to directly understand what the main focus of the article is. We truly believe that such a passage is usual and adequate for every review article. We hope you understand and agree with that.

  • Second point: Since no article has reviewed this literature topic until now, by writing this paper we aimed to sensitize the scientific world and especially the treating physicians to conduct further studies on this area. In the past, similar attempts have been done with antibiotic-loaded bone cement, leading to increased research about the elution of different antibiotic agents in vitro or in vivo. This review work is the first of its kind and therefore significant for every one who would like to start any study on this area.

  • Third point: We agree with this point. The fungus shown was isolated from a PJI case. Therefore, we have changed figure 1 by adding the X-rays of this case to the picture of the Candida spp..

  • Fourth pount: Thank you for your comment. We have added a new paragraph under section 3.2 explaining the role of bone cement in the treatment of hip and knee PJIs (under 3.2., lines 119-133)

  • Fifth point: as we mentioned in our last response, we have used the journal’s template for writing the manuscript. The order of the sections is defined by the journal and not us. Although we understand your point, we cannot subjectively change the format of the template if it is set by the journal in that way.

  • Sixth point: Thank you for this comment and we understand your point. As we also mentioned in our last response, we could cite several reviews which have a used a single database and are published, too. We understand that the scientific contribution might have been better if more databases had been searched through. For that reason, we have pointed this out in our limitation paragraph in the end of the Discussion. We will take this into consideration for our future research, but at this point we cannot change the whole manuscript and re-conduct the literature search. We hope that you understand this.

Because serious flaws in the current article make it dangerous to the scientific community and its low-quality means that this article should be rejected.

Response: Due to the afore mentioned explanations and the new changes of/ additions in our manuscript, we still believe that such a review contributes new knowledge to the present literature. With due respect, we find your term „dangerous to the scientific community“ not appropriate. We hope that after reading our comments you will change your mind about the decision of our paper.

With best regards

The authors

Reviewer 3 Report

Considering the changes made, I consider that the article meets the necessary conditions.

Author Response

Dear Reviewer, thank you again for reviewing our work and the kind reply.

The authors